# Refined Feasibility Testing of an 8-Week Sport and Physical Activity Intervention in a Rural Middle School

**DOI:** 10.3390/ijerph21070913

**Published:** 2024-07-12

**Authors:** Janette M. Watkins, Janelle M. Goss, McKenna G. Major, Megan M. Kwaiser, Andrew M. Medellin, James M. Hobson, Vanessa M. Martinez Kercher, Kyle A. Kercher

**Affiliations:** 1Department of Kinesiology, School of Public Health-Bloomington, Indiana University, Bloomington, IN 47405, USA; janhynes@iu.edu (J.M.W.); jangoss@iu.edu (J.M.G.); mkwaiser@iu.edu (M.M.K.); 2Program in Neuroscience, College of Arts and Sciences, Indiana University, Bloomington, IN 47405, USA; 3School of Medicine, Indiana University, Bloomington, IN 47405, USA; mckmajor@iu.edu; 4Department of Epidemiology & Biostatistics, School of Public Health-Bloomington, Indiana University, Bloomington, IN 47405, USA; ammedell@iu.edu; 5White River Valley Middle School, Lyons, IN 47443, USA; mhobson@wrv.k12.in.us; 6Department of Health & Wellness Design, School of Public Health-Bloomington, Indiana University, Bloomington, IN 47405, USA; vkercher@iu.edu

**Keywords:** strength training, feasibility testing, sport-based youth development, public health

## Abstract

This study examines how the 8-week Hoosier Sport program impacts cardiovascular disease (CVD) risks by promoting physical activity (PA) among rural, low-income children. Using a human-centered participatory co-design approach, the program aimed to increase PA levels (e.g., total PA, daily steps) in at-risk children. The present study explored the feasibility of the intervention as well as physiological and psychological changes across the intervention using a hybrid type 2 design (a model that evaluates both the effectiveness of an intervention and its implementation in real-world settings). Favorable feasibility indicators like attendance, acceptability, and compliance, with a 23.3% recruitment rate and 94.3% retention rate, were observed. Moreover, participants attended over 80% of sessions across the 8 weeks. Accelerometers (AX3) tracked daily steps and total PA for 7 days before and after the intervention, revealing increased PA levels throughout. At post-intervention, notable improvements were observed in psychological factors such as autonomy, social competence, and global self-worth. This study highlights the importance of tailored PA interventions in schools, emphasizing their potential to improve PA levels among rural, low-income children.

## 1. Introduction

Cardiovascular disease (CVD) is the leading cause of mortality for both men and women [1]. Physical inactivity is a major modifiable risk factor for CVD that can be altered through lifestyle and behavioral changes, especially in childhood. Only 20% of children in the U.S. meet the U.S. Department of Health and Human Services’ recommended guidelines for physical activity (PA) [2]. This lack of PA in children can lead to poor cardiometabolic health, obesity, and an increased risk for CVD in the future [3,4]. Due to behavioral risk factors and physical changes that can begin in childhood, interventions aimed at children are important for addressing physical inactivity and CVD.

Furthermore, children from rural, low-socioeconomic areas are disproportionately affected and face unique barriers to PA [5]. Many barriers stem from social determinants of health, such as economic inequality, limited resources, structural and environmental differences, and a lack of social support from family and peers [6]. For families and children of low socioeconomic status, financial strains can limit transportation [7] and involvement in more costly physical activities such as sports programs [6]. Additionally, environmental and structural factors like inadequate access to parks, playgrounds, and community centers can be a large barrier to PA for children in rural areas [8,9]. This lack of access can be further exacerbated based on the safety of the areas in which these community spaces are located [6]. In schools, structural factors include a lack of staffing resources or training for staff administering PA lessons [10]. Social barriers also exist for children, including stigma, poor parental support, and a lack of social networks such as friends and siblings [6,10]. Recognizing and mitigating these barriers is crucial in creating and implementing tailored interventions to increase PA in rural communities.

The development and testing of multilevel physical activity interventions is a priority for leading national public health agencies (i.e., National Institutes of Health) [11,12]. Further, there is a need for more campus-community partnerships to create trusting relationships and sustainable programming [11]. The Hoosier Sport program, in partnership with a remote, rural middle school, is a multi-level and multi-component intervention that takes a sport-based youth development approach with a primary focus on increasing PA levels in youth. A recent systematic review [12] found that sport-based youth development literature has primarily focused on assessing the cognitive, affective, social, and lifestyle domains, and there is an opportunity for additional emphasis on the physical domain (e.g., PA levels), Sport-based youth development is designed to use sport as the “hook” to engage children in holistic developmental programming [13]. In addition to providing recurring enhanced physical education classes, the intervention provides educational components and tools for empowering children to be physically active outside of school. PE classes across the U.S. only encompass around 41 to 46 min of a child’s day [14,15], and only 8% of middle schools in the U.S. provide opportunities for PA each day [16]. Thus, strategies are needed that build health literacy [17] and intrinsic motivation [18] to potentially increase children’s PA levels outside the classroom as well. To address this gap, the Hoosier Sport intervention was created collaboratively with children and adults from the community in a previously published, human-centered participatory co-design process [19]. The intervention was also designed to align with the basic psychological needs mini-theory within the self-determination theory [19,20]. Semi-structured co-design sessions were designed to include questions targeting competence, autonomy, relatedness, and enjoyment [21]. The theory highlights basic psychological needs—competence, autonomy, and relatedness—that must be satisfied to reach optimal well-being. Enjoyment was also measured based on its support for long-term PA adherence in children and satisfaction of psychological needs [22,23]. Attainment of needs can vary by sociocultural setting but ultimately leads to improved health in multiple facets.

Due to the complex nature of PA, a biobehavioral intervention is one potentially effective approach to intervention development and testing [21]. The long-term goal of the present intervention is to improve both biological and behavioral factors to increase PA. Our physiological outcomes are measured using research-grade accelerometers and vitals such as heart rate and blood pressure. Accelerometers and other activity trackers can be used as either the intervention target itself (e.g., usage/awareness of step tracking) or as a tool for modifying other behaviors (e.g., school walk breaks, home exercise) [24]. In the present study, they serve as a direct assessment of physiological outcomes. They also allow for a more objective and accurate analysis of PA than self-reported measures alone [25]. By assessing behavioral changes, we can identify specific intervention components that are most effective in promoting PA. This allows for the refinement and optimization of the intervention, increasing its potential for scalability and long-term sustainability. Because there is a relationship between PA and psychological need satisfaction, interventions can also assess psychological well-being [26]. The assessment of physiological, behavioral, and psychological factors is essential for creating a transdisciplinary approach to intervention design. By considering psychological factors such as cognitive processes, emotional responses, and individual motivations alongside physiological and behavioral data, interventions can be tailored more comprehensively to address the complex needs of individuals or populations.

Therefore, we conducted a pilot 8-week prospective study in a cohort (*n* = 33) of rural middle school children. Our quasi-experimental design included an intervention group (*n* = 22) and a control group (*n* = 11) that followed the standard school curriculum. Our primary objective was to assess trial- and intervention-related feasibility indicators, specific to the children’s engagement and perceptions. We hypothesized that Hoosier Sport would be feasible, which is interpreted as “good” scores for feasibility indicators (median of 16/20). The secondary objective was to assess pre-to-post changes in accelerometer-assessed PA. From an exploratory standpoint, we assessed changes in physical literacy (i.e., the ability and confidence to move effectively and enjoyably across a range of physical activities), psychological needs, child self-perception (e.g., academic abilities, social skills, physical competence, body image, behavior, and overall self-esteem), and the relationship between basic psychological needs and PA outcomes. Our hypothesis was that participants in our intervention group would have a greater improvement in these outcomes as compared to the control group.

## 2. Materials and Methods

### 2.1. Conceptual Framework

Our intervention was designed to target three levels of influence from the National Institute of Minority Health and Health Disparities (NIMHD) Research Framework [27]. This framework provides a comprehensive approach to addressing health disparities by considering individual, interpersonal, and community factors. The intervention strategies were developed using a human-centered participatory co-design approach, which has been detailed in our previous work [19]. This method involved actively engaging participants in the design process to ensure the strategies were relevant and effective for the target population. Furthermore, the mediating constructs of our intervention were selected based on the psychological needs mini-theory from self-determination theory (SDT), which emphasizes the importance of autonomy, competence, and relatedness [20]. By addressing these fundamental psychological needs, our intervention aimed to foster intrinsic motivation and sustained engagement in PA among participants.

As an overview of the conceptual framework, sport participation involves engaging participants in various physical and play-based activities to promote active lifestyles. Leadership development focuses on building participants’ skills to lead and motivate others in PA. Social support provides a network of encouragement and assistance from peers and mentors. Empowering education delivers knowledge and tools to make informed decisions about PA and health. Lastly, PA take-home equipment and activities offer resources for participants to continue their activities outside the structured environment, ensuring sustainability and long-term engagement. These strategies collectively create a supportive and motivating framework that addresses multiple levels of influence on PA behavior. Figure 1 provides further details on the conceptual framework of Hoosier Sport.

Furthermore, the intervention uses the Obesity-Related Behavioral Intervention Trials (ORBIT) model, which acts as a framework for behavioral change while also attempting to improve the testing and translation of behavioral interventions [28].

The ORBIT model emphasizes spending sufficient time and effort on early stage intervention development and feasibility testing, while situating the study as a crucial step toward conducting efficacy and effectiveness trials. As interventions often prematurely scale up or do not make it to the scaling up stage at all, incremental approaches to intervention testing are highly recommended [28,29]. The present Hoosier Sport intervention was co-designed with under-resourced children and adults to address the barriers and facilitators of the group (ORBIT Phase Ia) [19] and implemented after refinement following a previous pilot study (ORBIT Phase IIa) [30]. Moreover, analyzing a study’s feasibility helps ensure that interventions have effective designs with the ability to scale up [31]. Thus, it is imperative to address trial- and intervention-related indicators such as acceptability, recruitment, retention, and cost. Our measurement of feasibility indicators is focused on the children’s perceived acceptability of the intervention as well as their compliance and attendance.

### 2.2. Design

The present study used a hybrid type 2 design, which allows for simultaneously assessing implementation feasibility indicators while evaluating information and clinical outcomes [32]. Hybrid type 2 design offers a framework that integrates qualitative and quantitative data, enabling a formative evaluation of both the intervention’s preliminary signals and an assessment of its implementation feasibility [32,33,34]. This model involves systematically collecting data on the primary and secondary outcomes that indicate whether the intervention is achieving its intended effects. For example, Hoosier Sport focuses on feasibility indicators (e.g., FIM) while also improving PA outcomes (e.g., 6MWT) to combat CVD risks.

### 2.3. Intervention

Hoosier Sport extended over an 8-week period, comprising strength training sessions in weeks 1–4 and basketball activities in weeks 5–8. The intervention took place in-person during PE classes for 6th–7th graders, occurring twice a week with each session lasting approximately 45 min. These sessions were conducted by research assistants and graduate students from Hoosier Sport. Prior to the intervention, all research assistants and graduate students completed a one-day workshop focused on practical skills and measurement techniques, followed by an online assessment requiring a 100% passing score, ensuring they were well-prepared to facilitate the sessions. Additionally, all research assistants and graduate students completed the Collaborative Institutional Training Initiative (CITI) program and were listed on the Institutional Review Board (IRB) protocol, confirming their training in ethical research practices and data handling. Furthermore, all Hoosier Sport team members underwent background checks conducted by the community-partner school prior to attending in-person sessions. Moreover, an incentive reward system was introduced to encourage participants to engage in at-home fitness activities. Students earned points by completing workouts at home and could redeem these points for various items such as toys (e.g., fidget spinners, pop-sockets) and sports equipment (e.g., basketballs, Nerf footballs).

The research team gathered data at three pivotal time points: week 1 (baseline assessment), week 4 (midway through the intervention), and week 8 (post-intervention evaluation), while simultaneously tracking feasibility metrics such as attendance and retention throughout the entire duration of the intervention. Figure 2 provides a visual summary of the key time points within the intervention timeline.

### 2.4. Sample Size and Sample

To determine the appropriate sample size for our study, we conducted power analyses using G*Power (Version 3.1.9.6) [35]. For the paired-sample *t*-tests, an a priori power analysis was performed with a specified moderate effect size (Cohen’s d = 0.5), a significance level of 0.05, and a desired power of 0.80. The analysis indicated that a minimum of 30 participants would be required to achieve adequate statistical power. This effect size was chosen to align with the scope of a feasibility study and to ensure that meaningful differences could be detected without overestimating the effect [36].

The children in the study were students attending a rural middle school. To be eligible for the study, participants had to meet the following criteria: (1) they were in the 6th or 7th grade, and (2) they were enrolled in a PE class. Participants were excluded if they were advised by the PAR-Q+ or a healthcare provider that engaging in physical activity would be unsafe for them. The demographics of the included participants are shown in Table 1. The Indiana University Institutional Review Board approved the study protocol (#18784).

### 2.5. Measures

#### 2.5.1. Trial-Related Feasibility Indicators

The study assessed two trial-related feasibility indicators: recruitment capability and retention. Recruitment capability was measured by the number and percentage of students successfully recruited into the study out of the total student population in the school. Retention was measured as the percentage of students that remained in the study and attended post-data collection.

#### 2.5.2. Intervention-Related Feasibility Indicators

The study assessed the feasibility of the intervention through various indicators, including attendance, acceptability, appropriateness, overall feasibility, cost, and compliance. Specifically, adapted for children, the 4-item measures AIM (Acceptability of Intervention Measure), IAM (Intervention Appropriateness Measure), and FIM (Feasibility of Intervention Measure), were used. A Likert scale ranging from “completely disagree” to “completely agree” was utilized to gauge responses for AIM. Summary scores for each measure were calculated using means/medians and standard deviations. Additionally, participant attendance was evaluated to measure compliance, and at-home fitness scores were used to monitor participant engagement in intervention activities conducted outside of the in-person sessions.

The cost was calculated at the end of the intervention to include purchasing Axivity AX3 accelerometers, participant incentive gift cards, transportation of the research team to the rural school, and sports equipment to deliver the intervention.

#### 2.5.3. Physiological Assessment

This study deployed Axivity AX3 accelerometers (Axivity, Newcastle upon Tyne, UK), which have demonstrated high reliability in measuring sedentary and PA levels in children [37,38]. The device is a small, lightweight triaxial accelerometer that measures acceleration along three axes (vertical, anterior-posterior, and medio-lateral) at a sampling frequency of 100 Hz with a range of +8 g to −8 g. Indeed, AX3 models can differentiate between six distinct activity classes with exceptionally high accuracy in children (97.3%) [39]. Moreover, the device is valid for calculating walking steps [38,40]. The AX3 was used to assess PA levels and daily steps at pre- and post-intervention.

Participants received comprehensive instructions on how to properly wear the accelerometer. They were directed to wear the device on their non-dominant wrist using an adjustable wristband supplied by the researchers. To ensure the device was oriented correctly, stickers were applied, providing a visual cue for proper wear. Each accelerometer was initialized prior to use to guarantee accurate data collection. Participants were instructed to maintain their usual daily activities and avoid changing their PA patterns during the data collection period. The devices were programmed to start collecting data the day after distribution to minimize novelty bias, with data collection lasting for seven days.

For cardiovascular health measures, blood pressure and heart rate were assessed pre- and post-60 s squat test and 6 min walk test (6MWT) [41]. The participants underwent a comprehensive series of assessments and tests as part of the intervention program. These assessments were conducted at specific time points to monitor changes in physical performance and perceived exertion throughout the intervention period. At the beginning of the intervention (week 0), week 4, and week 8, participants completed a sequence of tests. First, they performed the 6 min walk test (6MWT), which involved walking laps in a school gym while a researcher counted the laps completed. Before and after the 6MWT, participants had their blood pressure (BP) and heart rate (HR) measured, and they rated their perceived exertion (RPE) on a scale of 1 to 10. After a 5 min rest period following the 6MWT, participants completed the max plank test, where they held a plank position for as long as possible.

#### 2.5.4. Psychological Outcomes

The Basic Psychological Needs Satisfaction and Frustration Scale (BPNSFS) was used to assess psychological outcomes at baseline and post-intervention. The BPNSFS has demonstrated adequate internal consistency with Cronbach’s alpha coefficients of global basic psychological need satisfaction of 0.9, 0.80 for autonomy satisfaction, 0.81 for competence satisfaction, and 0.81 for relatedness satisfaction [42,43].

The Self-Perception Profile for Children (SPPC) [44] assesses children’s self-concept across six specific domains or sub-scales. These include Scholastic Competence, which evaluates academic abilities; Social Competence, measuring social skills and interactions; Athletic Competence, assessing physical abilities and sports performance; Physical Appearance, focusing on body image perceptions; Behavioral Conduct, examining self-discipline and rule-following; and Global Self-Worth, providing an overall assessment of self-esteem across all domains. The SPPC has shown high reliability in children and adolescent samples (α = 0.83–0.95) [45].

### 2.6. Procedure

Children from a rural middle school were recruited through flyers distributed during school lunch hours and emails sent to parents. The parents of interested children were contacted via email, and a parental call was scheduled to explain the study and obtain verbal consent. Additionally, parents completed the Physical Activity Readiness Questionnaire (PAR-Q) [46] to assess their child’s eligibility and provide written consent via an online survey. Once parents provided consent, children were identified and excused from class to undergo data collection in a designated quiet area within the school gym before the first day of the intervention. During this baseline data collection, children were asked to provide written assent in person with the research team, giving them the option to participate. After obtaining assent, children completed survey measures and fitness assessments (as described in Section 2). These assessments were conducted under controlled conditions in the same quiet area within the gym to ensure consistency and accuracy. Trained research assistants, who had completed specialized training in administering the assessments and handling sensitive data, guided the children through each step of the process, providing support. Children completed the survey data collection on their tablets, provided by the school, which included the BPNSFS, CAPL-2, SPPC, and demographic information. Following the surveys, participants completed fitness assessments in the following order: the 6-minute walk test (6MWT) and the plank test. Data collection took approximately 25 min for each participant. The same data collection process was repeated in week 8, following the conclusion of the intervention.

### 2.7. Data Analysis

The data were analyzed using the latest version of R Studio (version 2024.04.2+764) [47]. The analysis of pre- to post-intervention results involved a thorough examination of key health measures, including HR, BP, plank scores, outcomes from the 6MWT, daily steps, and total PA in 7 days. Paired-sample *t*-tests were employed to determine the significance of within-subject differences between pre- and post-intervention measurements for each parameter. Specifically, *t*-tests were conducted to compare HR and BP values, Plank scores, distances covered during the 6MWT, daily steps, and total PA before and after the intervention.

Axivity data were downloaded using Open Movement Raw Axivity AX3 .csv. Accelerometer files were processed in R (version 2024.04.2+764) (using the GGIR package) [48]. During processing, GGIR auto-calibrates the signal using local gravity as a reference, identifies abnormally high values, detects non-wear periods, and calculates the magnitude of the acceleration corrected for gravity. Files were excluded if the post-calibration error exceeded 0.02 g or if the wear time was less than 16 h during the 24 h period. The default method for non-wear time was used, with invalid data imputed based on the average acceleration at similar times on other days of the week. Total non-wear time was converted to days to assess compliance across the 7-day data collection period. Additionally, PA intensity levels and duration were determined using summary reports provided by GGIR. Finally, R code was developed to calculate average daily steps, as this functionality is not provided by the GGIR package. The magnitude of acceleration was computed as the Euclidean norm of the three axes: $\sqrt{X^{2}+Y^{2}+Z^{2}}$. To set an appropriate threshold for step detection, the sensitivity of the accelerometer was considered, defined as 8 × 9.81/1000 m/s^2^ per g. A threshold value in terms of g was converted to m/s^2^ and adjusted for the accelerometer’s sensitivity and the sampling frequency of 100 Hz. Steps were detected by identifying instances where the magnitude of acceleration exceeded the threshold. The total number of steps was then calculated by summing the detected steps.

## 3. Results

### 3.1. Trial-Related Feasibility Indicators

We successfully recruited 23.3% of the total middle school enrollment for the study, enrolling 35 out of 150 students. Of those, we retained 33 participants, achieving a retention rate of 94.3%.

### 3.2. Intervention-Related Feasibility Indicators

Our main goal was to evaluate the feasibility indicators of the study, focusing primarily on children’s engagement and perceptions. Attendance during the study was strong: participants attended 80.15% of the sessions in the first 4-week segment focused on strength training, and 88.45% of the sessions in the second 4-week segment focused on basketball. Table 2 shows the feasibility indicators for both the control and test groups.

To evaluate compliance, we calculated the wear time and return rate for the AX3 devices. For pre-intervention, the control group wore the AX3s for an average of 4.5 days (*SD* = 2.5) out of the assigned 7 days, while the test group averaged 5.5 days (*SD* = 2.4). At post-intervention, the control group’s average wear time increased to 4.5 days (*SD* = 2.5), and the test group increased to 5.8 days (*SD* = 1.8). A paired *t*-test revealed no significant change in compliance for the control group (*p* = 0.051) or for the test group (*p* = 0.050). Regarding the return rate, the pre-intervention rate was 91.67% for the control group and 95.23% for the test group. At post-intervention, the return rate dropped to 83.34% for the control group and 85.71% for the test group. Neither group showed a significant change in return rate (*p* > 0.05).

### 3.3. Cost

The cost to conduct the intervention was USD 12,050. Costs included purchasing Axivity AX3 accelerometers, participant incentive gift cards, mileage for transportation of the research team to the rural school on 22 days (intervention, recruitment, data collection), incentives for the points system, and sports equipment to deliver the intervention (i.e., new strength training equipment and basketballs), but did not include research team salaries.

### 3.4. Physiological and Psychological Outcomes

Our secondary aim was to evaluate PA using physiological and psychological measures. Using paired *t*-tests, it was found that there was a significant increase in the test group for the sense of autonomy (*p* = 0.023) post-intervention compared to pre-intervention, with a mean increase of 2.779 (*SD* = 4.21). Similarly, social competence significantly increased for the test group *(p* = 0.005) at post-intervention compared to pre-intervention, with a mean increase of 2.476 (*SD* = 2.39). Additionally, the global score exhibited a significant increase for the test group (*p* = 0.028) post-intervention compared to pre-intervention, with a mean increase of 1.381 (*SD* = 1.77). No such changes were observed in the control group. Finally, for physical literacy, the control group mean scores were 6.54 (*SD* = 1.96) at pre-intervention and 6.27 (*SD* = 4.02) at post-intervention, with no significant change (*p* = 0.412). For the test group, mean CAPL-2 scores were 5.13 (*SD* = 2.12) and 5.95 (*SD* = 3.44) at post-intervention. No significant change was observed in physical literacy for the test group (*p* = 0.123). Table 3 shows *t*-test results.

To assess PA, daily average steps and total PA (i.e., low, moderate, and vigorous PA) were calculated from AX3 wear. Table 4 shows all PA levels. To explore differences among groups, two sample *t*-tests were conducted. No differences were seen between the control and test groups at pre- or post-intervention in total PA or average daily steps. However, both the test and control groups showed a significant increase in total PA and steps from pre- to post-intervention. Table 5 shows all *t*-test results.

### 3.5. Physiological and Psychological Relationships

Our third objective aimed to assess potential associations between physiological and psychological outcomes. Linear regression analyses were employed for this purpose. The regressions were carried out using pre-intervention psychological subscales, such as autonomy, competence, and relatedness, along with the composite score representing basic psychological needs, as the independent variables. The dependent variables used were total PA, daily average steps, plank score, and 6MWT score. The analyses did reveal a statistically significant relationship between pre-intervention autonomy and post-intervention plank score (*p* = 0.005). Baseline psychological needs did not predict physiological outcomes at post-intervention. For a comprehensive overview, please refer to Appendix A, which presents all regression results.

## 4. Discussion

The present study took place within an established campus-community partnership between a Midwestern university and an under-resourced rural school district. The primary objective was to assess trial- and intervention-related feasibility indicators in a cohort of 33 rural middle school children. Secondarily, we aimed to assess the feasibility of Hoosier Sport and gather preliminary data on potential behavioral (e.g., daily steps, total PA) and psychological (e.g., basic psychological needs, global self-worth, etc.) changes. There were four key findings from this study. First, in line with our primary objective and hypotheses, trial- and intervention-related feasibility outcomes were above a priori thresholds for progression criteria. Second, accelerometer-derived PA levels increased in both the test and control groups from pre- to post-intervention. Third, from an exploratory standpoint, subscales of global self-worth and social competence in the self-perception profile for children increased from pre- to post-intervention. Fourth, and also exploratory, the basic psychological need for autonomy increased from pre- to post-intervention in the test group. This study provides valuable preliminary data to inform subsequent trials and refinements in line with the ORBIT model.

The first key finding was that the intervention met a priori thresholds for progression criteria in eight trial- and intervention-related feasibility indicators. Despite their importance to future clinical trial scaling decisions, many trial- and intervention-related feasibility indicators are underreported in research [33]. In a recent historical scoping review of prominent feasibility indicators reported in pilot studies, feasibility indicators were reported at the following frequencies (highest reporting to lowest): retention (99% of studies), recruitment (71%), quantitative acceptability (37%), attendance (33%), compliance (31%), qualitative acceptability (26%), treatment fidelity (14%), and cost (4%). For trial-related indicators in the present study, recruitment (initial *n* = 35) and retention (94%) targets were met. Our recruitment capability expanded in this study compared to the initial pilot intervention as the research team had increased access to students (e.g., during lunch time, physical education, health class) [49]. School administrators were highly supportive of the campus-community partnership and corresponding recruitment efforts. In terms of retention, the study had 94% participant retention. The only participants lost (*n* = 2) were due to moving out of the school district.

There were also promising results for intervention-related feasibility indicators (i.e., acceptability, appropriateness, attendance, compliance, and cost). AIM, IAM, and FIM, assessing children’s perceptions of the acceptability, appropriateness, and feasibility of Hoosier Sport, respectively, each achieved over our target thresholds of 80%. Since research has found that less than 1% of research involving children includes children’s opinions in the research process, these high scores were critically important to moving forward with subsequent trials. Had the scores been below target thresholds, it would have been important to investigate potential sources of low scores and generate new solutions to align the intervention with children’s interests. Cost is one of the most important feasibility indicators, yet it is rarely discussed in research (only 4% of the time) [33]. Perhaps cost is typically omitted because researchers are generally not trained as diligently in the best practices of project management as business professionals. Another potential reason could be that journal reviewers do not encourage the sharing of cost information sufficiently during the peer-review process. Regardless, the cost to deliver this intervention was USD 12,050, excluding personnel costs, as those are highly variable between organizational contexts. Transparency with this estimate will help inform projections for future trials, including sample size, duration, and curricular planning, as well as helping researchers looking to conduct pilot studies in similar under-resourced rural school settings. Giving greater attention to costs associated with delivering interventions may help researchers be better informed to make important intervention design decisions, such as navigating the level of rigor in feasible PA assessment (e.g., accelerometer vs. self-report measures). These promising feasibility findings will help inform the next stages of testing in line with the ORBIT model.

The second finding indicated that both the control and test groups showed improvements in daily average steps and total PA. Notably, other research on school-based PA interventions in youth has also observed improvements in the control groups [47,48,49]. These changes in the control groups may be attributed to increased awareness of PA within the school environment due to the intervention or program. This heightened awareness likely influenced the behavior of the control group, even though they were not directly participating in the program. While the changes observed in the control group make it challenging to draw definitive conclusions about the intervention’s effectiveness, it is encouraging that a school-based intervention can potentially positively influence children outside of the program. Moreover, the present study demonstrates the feasibility of school-based PA interventions in boosting both daily steps and overall activity levels. This finding is supported by other empirical research [50], suggesting that schools are an ideal setting for promoting healthy lifestyles. Additionally, such programs can help to address health disparities/inequalities in certain subgroups (e.g., low SES, bodyweight status), such as by increasing PA opportunities for children who would otherwise not have access to them [51]. Intervention designs should continue to focus on implementation within schools, emphasizing broad engagement and creating an environment that fosters PA for all students.

The third key finding was an increase in self-perception for students in the intervention, as measured by the Self-Perception Profile for Children. Specifically, the domains of social competence and global self-worth improved from pre- to post-intervention. A recent systematic review has demonstrated that PA interventions can enhance behavioral and cognitive-perceptive aspects of social competence in youth [52]. Indeed, through team sports and group exercises, children engage in peer interactions, fostering teamwork and communication skills essential for effective collaboration [53]. Additionally, taking on leadership roles within these programs, such as being a team captain or group leader, enables youth to develop decision-making abilities and the capacity to motivate and guide others [54], contributing to their overall social competence. Furthermore, these programs create inclusive environments where participants from diverse backgrounds come together, promoting acceptance, empathy, and respect [55,56]. This inclusive atmosphere not only encourages positive social attitudes but also provides valuable experiences in navigating differences and fostering a sense of belonging, all of which are integral aspects of social competence development in youth. Additionally, there was an increase in global self-worth seen from pre- to post-intervention. A longitudinal study has explored the effects of PA interventions on global self-worth in youth and found that increased levels of PA are correlated with increased levels of self-worth (and this trend did not vary across the 4-year study) [57]. Other work exploring this outcome suggests that improvements in global self-worth from PA may be due to enhanced physical self-esteem [58]. Therefore, PA interventions may aim not only for physical health but also for the promotion of positive self-esteem and overall well-being. By prioritizing PA in youth programs, a supportive environment can be created that contributes to the holistic development of young individuals, equipping them with the confidence and self-worth necessary for success in various aspects of their lives.

The fourth key finding was that the basic psychological need of autonomy increased from pre- to post-intervention in the test group. Hoosier Sport was designed in part to satisfy the psychological needs of children, in line with self-determination theory [22]; thus, this preliminary clinical signal is encouraging for children striving towards optimal well-being. Encouraging a sense of autonomy was built into both the strength training and basketball curriculum by providing children with the opportunity to choose between physical activities (e.g., drills, games, partners/teammates) as well as asking them what they would like to do at times when there was extra time in the lesson plans. This finding aligns closely with the Aspen Institutes’ Project Play 8 Plays Framework [59], which suggests one of the keys to improving youth sport participation in the United States is to ask children what they want more often. This finding is particularly interesting as psychological needs are relatively stable across short time periods (i.e., weeks) and do not typically see significant changes unless they are of greater duration (more longitudinal) [60]. Furthermore, the present study aimed to assess if baseline basic psychological needs predicted PA outcomes (e.g., 6MWT, plank test, daily steps, total PA). It was seen that baseline autonomy predicted post-intervention plank times. However, there were no other significant relationships between baseline and post-intervention performance. Moving forward, including basic psychological needs in more longitudinal work may help to better explore this idea of how psychological needs predict behaviors in the context of PA.

There were limitations in this study. First, the control group was smaller than the test group, which may have impacted the statistical power and generalizability of the results. While it was beneficial that Hoosier Sport occurred during PE classes, leading to higher attendance and accessibility, nearly all sixth graders and most seventh graders are required to take PE, making it challenging to identify eligible control students. Additionally, although the participants generally complied with wearing the accelerometers for at least four days, the data were often incomplete for the full 7-day collection period. This resulted in potential data gaps that may have affected the accuracy and reliability of the activity measurements. Consequently, significant changes might not be observed unless the study duration is extended to capture more longitudinal data and better incentives or systems are implemented to increase compliance.

## 5. Conclusions

This study underscores the feasibility and potential impact of the Hoosier Sport intervention on enhancing both psychological and physiological health in a rural middle school setting. The intervention successfully met key feasibility indicators, demonstrating its viability for future, larger-scale implementations. Notably, the program led to increased PA levels in both intervention and control groups, highlighting the potential of school-based interventions in promoting healthier lifestyles for the entire student body. Moreover, significant improvements in self-perception, including social competence and global self-worth, as well as a heightened sense of autonomy, suggest that such programs can contribute to the psychological well-being of students. These findings advocate for the integration of structured PA programs within school curricula to support the holistic development of children. Despite limitations such as group size imbalances and incomplete data collection, this study provides a strong foundation for refining and expanding school-based interventions aimed at improving both the physical and psychological health of students in under-resourced areas.

## Figures and Tables

**Figure 1 ijerph-21-00913-f001:**
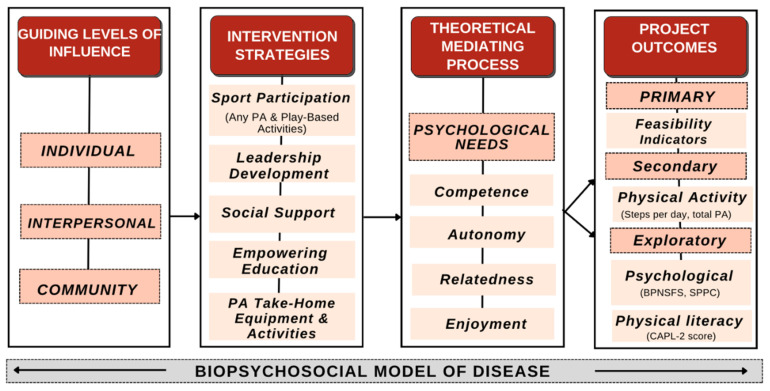
Hoosier Sport conceptual framework. Each section represents key components of the framework, with accompanying details on the specific components being applied or implemented. PA = physical activity; BPNSFS = Basic Psychological Needs Satisfaction and Frustration Scale; SPPC = Self-Perception Profile for Children; CAPL-2 = Canadian Assessment of Physical Literacy Second edition.

**Figure 2 ijerph-21-00913-f002:**
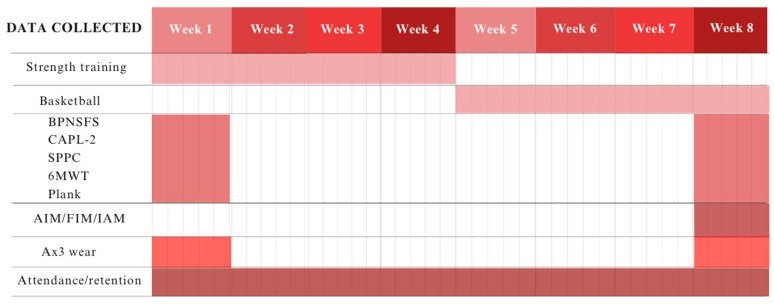
Timeline of data collection. Columns represent the time points of the intervention, while rows represent the components of the study being implemented. Shading within the figure indicates whether each component was implemented or not. BPNSFS = Basic Psychological Needs Satisfaction and Frustration Scale; CAPL-2 = Canadian Assessment of Physical Literacy (Second Edition); SPPC = Self-Perception Profile for Children; 6MWT = 6 min walk test.

**Table 1 ijerph-21-00913-t001:** Demographics.

	Test Group	Control Group
Gender (*n*)		
Female	14	5
Male	10	6
Age (M, *SD*)		
Female	12.4 (0.5)	12.9 (0.3)
Male	12.7 (0.4)	12.7 (0.3)
CAPL-2 Score (M, *SD*)		
Female	4.18 (2.08)	6.45 (3.69)
Male	6.09 (1.75)	5.88 (2.97)

**Table 2 ijerph-21-00913-t002:** Feasibility indicators (*n* = 33).

Variable	Test Mean	Test Median	Test *SD*	Control Mean	Control Median	Control *SD*
FIM	17.42	18	2.63	16.63	17	4.12
AIM	16.85	17	3.69	17.72	19	2.28
IAM	17.23	18	2.62	16.09	20	5.59

Note: FIM = Feasibility of Intervention Measure. AIM = Acceptability of Intervention Measure. IAM = Intervention Appropriateness Measure.

**Table 3 ijerph-21-00913-t003:** *t*-Test results for pre vs. post.

Variable	Test Pre-Post Mean Difference	Test *p*-Value	Control Pre-Post Mean Difference	Control *p*-Value
Plank	−1.44	0.891	15.31	0.395
6MWT	−14.71	0.467	−26.55	0.168
Resting HR	−3.17	0.407	3.27	0.371
Autonomy	2.78	0.023 *	2.00	0.300
Competence	−0.58	0.670	−0.73	0.757
Relatedness	0.90	0.536	1.55	0.516
Athletic Competence	−1.14	0.130	−0.82	0.426
Social Competence	2.48	0.005 *	−1.00	0.542
Global Self-Worth	1.38	0.028 *	−1.73	0.132
CAPL-2	0.816	0.123	0.27	0.412

Note: Autonomy, competence, and relatedness are subscales of the Basic Psychological Needs Satisfaction and Frustration Scale (BPNSFS). Athletic competence, social competence, and global self-worth are subscales of the Self-Perception Profile for Children (SPPC). 6MWT = 6 min walk test; HR = heart rate; CAPL-2 = Canadian Assessment of Physical Literacy (version 2). * indicates a significant *p*-value.

**Table 4 ijerph-21-00913-t004:** Physical activity levels.

Variables	Control Mean (*SD*)	Test Mean (*SD*)
Pre-Intervention		
Avg. Daily Steps	2071.50 (2172.79)	4213.71 (4954.87)
Total PA (min)	127.41 (72.58)	181.011 (122.391)
Post-Intervention		
Avg. Daily Steps	2681.05 (2739.53)	5898.19 (8807.48)
Total PA (min)	294.27 (201.52)	324.23 (223.11)

**Table 5 ijerph-21-00913-t005:** *t*-Test results of physical activity levels in control vs. test groups.

Variables and Time	*t*-Score	95% CI	*p*-Value
Control vs. Test			
Pre-Intervention			
Avg. Daily Steps	1.612	[−4099.308, 3409.243]	0.118
Total PA (min)	1.637	[−20.499, 128.327]	0.115
Post-Intervention			
Avg. Daily Steps	1.35	[−1333.400, 5384.334]	0.192
Total PA (min)	0.741	[−153.987, 212.575]	0.741
Test Group			
Avg. Daily Steps	2.484	[−1017.728, 7814.223]	0.025
Total PA (min)	2.293	[−272.607, 130.840]	0.031
Control Group			
Avg. Daily Steps	2.36	[−4582.974, 3674.117]	0.041
Total PA (min)	2.702	[−326.006, 123.049]	0.026

## Data Availability

The datasets used and/or analyzed during the current study are available from the corresponding author on reasonable request.

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
