# Peer review of "Refined Feasibility Testing of an 8-Week Sport and Physical Activity Intervention in a Rural Middle School"

_ijerph, 2024, doi:10.3390/ijerph21070913_

Round 1

Reviewer 1 Report

Comments and Suggestions for Authors

Author Response

Please see the attached word document for all author responses. Thank you for taking the time to review our work and make it better. 

Reviewer 2 Report

Comments and Suggestions for Authors

The article is of interest but needs some changes to be made before it can be considered for publication.

In the abstract they should indicate what they mean by hybrid program type 2.
The introduction should be reworded. Methodological aspects of the work done should not be established, but a review and background on this topic should be made.
Methods:
- The authors should define in greater detail, what a type 2 hybrid program consists of.
- The participants should be better defined. Inclusion and exclusion criteria should also be included. The exercise habits and sports history of the participants should also be included.  In addition, the sample size calculation should be justified.
- Figures should contain a figure caption with all the abbreviations used.
- In addition, the authors should state under what conditions and by whom the measurements were taken. What experience did he/she have to carry them out.
- Regarding the control of contaminating variables, has a control of the exercise practice, diet etc, of the participants outside the program been carried out?

Results
- I think that point 3.3 does not need to be included in this manuscript.
- The ethics committee that approved this research is not indicated, as well as its registration number.

Author Response

Please see the attached word document for author responses to reviewer comment. Thank you for taking the time to review our work and make it better. 

Round 2

Reviewer 2 Report

Comments and Suggestions for Authors

the authors still do not answer what training or qualifications the persons responsible for each measurement had. As well as under what conditions of location etc. they were performed, and what experience or training the evaluators who carried them out had.

Author Response

Thank you for taking the time to provide a second review. Please see the attached document for the author responses. 
